# Strategic Choice and Implementation of Workplace Wellness Programs in the United States

**DOI:** 10.3390/healthcare10071216

**Published:** 2022-06-29

**Authors:** Marc Weinstein

**Affiliations:** Department of Global, Leadership and Management, College of Business, Florida International University, Miami, FL 33199, USA; weinstei@fiu.edu

**Keywords:** strategic choice, wellness programs, health promotion

## Abstract

Despite widespread discussion and public policy support for workplace wellness programs in the United States, their diffusion has been slow. Using data from the 2017 Workplace Health Administration Survey, this paper explored the importance of establishment characteristics, unionization, and strategic choice in the adoption of workplace health initiatives and employee participation in these programs. An ordinary least squares analysis revealed that unionization (β = 1.59, 95% CI = 1.20–1.97, *p* < 0.001) and management support (β = 1.67, 95% CI = 1.25–2.10, *p* < 0.001) were the strongest predictors of the number of programs adopted by an establishment. In logistic regression analyses of nine workplace wellness programs, it was also found that unionization and management were the strongest predictors of the adoption of these programs. Management support was also correlated with employee participation of in nutrition (OR = 2.66, 95% CI = 1.23–5.71, *p* < 0.05) and obesity programs (OR = 3.66, 95% CI = 1.03–12.97, *p* < 0.05).

## 1. Introduction

### 1.1. Workplace Health Promotion in the United States

In the United States, there has been a longstanding interest in workplace health programs (WHPs). In the years following World War II, workplace programs to address alcoholism were established and later broadened and institutionalized as employee assistance programs [1]. By the 1980s, many companies extended workplace initiatives to address a broad range of preventable diseases related to poor diet and lack of exercise [2]. In the last three decades, the factors driving interest in WHPs have endured. Obesity rates in the US have steadily climbed, with nearly one-third of American adults having a body mass index (BMI) higher than 30 [3].

The systematic investigation of WHPs has followed these trends. In 1985, the US Center for Disease Control (CDC) commissioned the first Workplace Health Administration Survey (WHA) to gauge how behavioral health issues were being addressed in the workplace. By 1990, the CDC issued the Healthy People report that identified elements of a comprehensive WHP to be (1) health education programs, (2) supportive social and physical work environment, (3) integration into the organization, (4) linkages to related programs, and (5) health screening follow-up [4].

In the complex patchwork of healthcare coverage in the United States, employer-sponsored health insurance covers more than 50 percent of the non-elderly population [5]. The Affordable Care Act (2010) further strengthened this arrangement by mandating insurance for full-time employees in companies with more than 100 employees. Given the centrality of work for many Americans [6], long work hours [7], and the potential mutual benefit workplace health initiatives offer individuals and companies in terms of enhanced health and lower healthcare costs, the workplace has become a logical locale for the promotion of behavioral health [8].

The Affordable Care Act reinforces this by providing employers the opportunity to offer deductible incentives to encourage employee participation in WHP. The evolution of self-insurance programs for employers provides further opportunities for employers to have lower healthcare costs through improving overall employee health [9]. Further, there is increasing evidence that, regardless of their participation, the promotion of WHP initiatives increases employee satisfaction [10]. Finally, although the COVID-19 pandemic of 2020–2021 may have placed WHPs on a momentary pause, the need for WHPs is likely to remain strong, as most Americans reported additional weight gain during this period [11].

Despite what appears to be numerous forces driving the adoption of WHP, their growth has been quite modest [12]. One factor that seems to mitigate employer interest in WHPs is the mixed data on program effectiveness. Early reports on WHPs oversold their effectiveness [13], with more rigorous studies indicating that most WHP initiatives typically engage individuals who are already pursuing healthy lifestyles [14]. Despite this, numerous companies have reported substantial success, and the role of behavioral health change on the incidence of various diseases is well-established [15,16].

### 1.2. Functional and Strategic Choice Perspectives

Given the history of WHPs in the United States, healthcare cost pressures, and the mutual benefits WHPs offer, their relatively slow diffusion is confounding. Drawing on the tradition of organizational studies, there are two broad theoretical frameworks to examine this. The first is a functional framework that seeks to explain variation in organization practices as a function of environmental factors [17]. The second perspective emphasizes human agency and the strategic decisions of organizational leaders [18].

In the area of WHP initiatives, a functional approach explores the relationship between the structural attributes of an enterprise and the adoption of WHPs. Thus, firm size has been a particular variable of interest, given the assumption that there are economies of scale in developing WHPs. Similarly, the industrial sector is also considered relevant, since some sectors in the US (e.g., public employees) have longer job tenure or cost advantages (e.g., healthcare) when introducing WHPs.

Studies exploring the adoption of WHPs within a functional framework include Linnan et al.’s 2004 analysis of the WHA in which the authors found lower adoption of the five key components of WHPs in smaller enterprises [19]. Linnan et al. replicated this finding in their analysis of the 2017 WHA [12]. Smogor et al. found lower adoption in smaller enterprises, though they emphasized a lack of knowledge of WHPs in this 1987 study [20]. Wilson et al. reached a similar conclusion in 1999, noting that small enterprises tended to focus on workplace hazards rather than general human wellness [21]. More recently, Weinstein and Cheddie highlighted industry differences [22], as did a recent analysis of the adoption of sleep enhancement programs [23].

Indeed, firm size and industry sector are logical factors to examine from a functional perspective that assumes cost reduction is the primary factor driving the adoption of WHP. However, given the nature of chronic diseases, even the most effective WHPs will likely only have long downstream returns. In an economic environment in which corporate leadership is often measured in terms of quarterly performance, the functional framework may be limited in explaining the pattern of adoption of WHPs.

An alternative approach would be to use a strategic choice framework [18]. This perspective incorporates the importance of human agency and organizational politics in decision making. In this perspective, top management has a sphere of autonomy to establish the priority for programs, regardless of initial data on effectiveness. In the case of WHPs, this strategic patience may indeed create a positive feedback loop. The strategic decision to launch and sustain these programs may provide them sufficient time to mature and reach a tipping point where they either directly contribute to a reduction in healthcare costs or prove to benefit the employment brand of the company.

## 2. Materials and Methods

### 2.1. Study Design and Methods

In this analysis, we used the data from the 2017 Workplace Health Administration Survey, a nationally representative, cross-sectional survey of establishments that asks a series of questions related to WHPs. The 2017 survey is the most recent of five, with earlier versions being administered in 1985, 1992, 1999, and 2004.

The 2017 sample was randomly drawn from the population of 2.5 million private and public establishments with at least 10 employees. Designed to provide a nationally representative sample of establishments, the sample was stratified based on industry (7 categories), size (7 categories), and CDC region (10 categories). Following Linnan et al., we collapsed the 750+ size category into the 500+ employee category, due to the relatively small number of firms in the two largest categories of establishments [12].

Trained interviewers initially contacted each worksite in the sample by phone to identify and recruit the person “most knowledgeable about employee health and safety at the worksite”. The interviewers then confirmed that the establishment had at least 10 employees and had been in operation for at least 12 months. Respondents then were invited to complete the survey by mail (4.9%), a telephone interview (8.6%), or via the Internet (86.6%) [12]. In 30.6 percent of establishments, this was a human resource manager, while in 3.8 percent, it was a wellness or health promotion professional; the remaining respondents were from a wide variety of job titles. Interviews averaged 40 min and were conducted between November 2016 and September 2017.

Table 1 provides a snapshot of the distribution of establishment in the sample by industry group and size. The size and industry groups presented in Table 1 are the same as the original categories of the WHA.

### 2.2. Measures

The measures in the analysis were derived from the 2017 WHA. The industry categories used in the analysis are the same as those used in the original WHA. This is true for firm size, except as noted above, where the two largest sized categories were collapsed. A dichotomous variable was created to indicate which establishments had a union density greater than 10 percent. A strategic choice variable to support WHPs was created from the yes/no question HP7A in the dataset: “Has senior leadership visibly committed to employee health and safe work environment?” This last question was only asked at the 1484 establishments (52.2 percent of the valid sample) that had at least one workplace health program in the last 12 months.

Three sets of dependent measures were used in this analysis: the number of WHPs, the existence of specific workplace health initiatives, and participation in a workplace health program. Regarding specific initiatives, respondents were asked if they had workplace health programs in the last 12 months in nine areas: (1) physical activity, (2) nutrition, (3) obesity, (4) tobacco abatement, (5) alcohol abuse, (6) lactation support, (7) musculoskeletal disease (MSD) programs, (8) stress management, and (9) sleep management. From these, the variable “number of programs” was created that ranged from 1 to 9. Table 2 provides data on the number and percentage of establishments that have at least one WHP initiative. Similar to Table 1, the number of firms and percentages are provided by establishment size and industry in categories defined by the WHA.

Data on employee participation were provided for seven of the nine WHP initiatives. Data for tobacco cessation were not collected, and the number of employees participating in lactation programs was too small for analysis. The variables on participation were derived from a four-level categorical variable. This variable was coded as 1 when at least 25 percent or more of the employees participated in a particular WHP and coded 0 for participation less than 25 percent.

### 2.3. Analysis

In this analysis, three approaches were used to explore the relative importance of firm size, industry, unionization status, and strategic choice in the adoption of WHPs. First, to analyze the number of programs, we utilized an ordinary least square regression. This analysis has four hierarchically presented models, with new independent constructs that are successively added, allowing an assessment of functional and strategic choice variables in explaining establishment-level adoption of WHPs. Second, we used logistic regression to explore the adoption of the nine specific WHPs about which the WHA inquired. Finally, we used logistic regression to explore the relationship between the same set of independent variables and employee participation for the seven types of WHPs for which we had data. In all models, firms with less than 25 employees were the referent group, used as the base of the model. For those models that included the industrial sector, the industrial category of agriculture, mining, utility, and construction was the base of the model.

## 3. Results

Table 3 lists the results of the ordinary least square used to model the number of programs in each establishment. Model 1A included variables on firm size, Model 1B added the industry variables, Model 1C added the variable on unionization status, and the fully specified model 1D included these variables and added the variable on senior leadership support. The referent group in the base of the models for establishment size was 10–24 employees; the referent group for the industry was the heavy industry group that included agriculture, mining, utilities, and construction. The F test for all models was statistically significate at *p* < 0.0001. The adjusted R squares for the base model that only included firm size was 0.067 and increased to 0.092 when industry variables were added in Model 1B. The adjusted R square further increased to 0.184 when the union variable was added (Model 1C) and increased to 0.234 when in the fully specified Model 1D.

In the fully specified Model 1D, establishments with between 25 and 49 employees were predicted to have a fewer number of workplace wellness initiatives (β = −0.45, 95% CI = −0.90–0, *p* < 0.05) than the smallest category of firms (10–24 employees) in the base of the model. This was also observed for establishments with 50–99 employees (β = −0.72, 95% CI = −1.23–−0.21, *p* < 0.01). At the other end of the size spectrum, establishments with more than 500 employees were positively associated with the number of programs (β = 1.73, 95% CI = 1.16–2.30, *p* < 0.001). With regard to industrial sector, retail (β = 0.63, 95% CI = 0.04–1.23, *p* < 0.05), public administration (β = 0.71, 95% CI = 0.11–1.31, *p* < 0.05), and hospitals (β = 0.80, 95% CI = 0.21–1.40, *p* < 0.01) were positively correlated with the number of programs. Establishments with union density of at least 10 percent were also predicted to have a higher number of programs (β = 1.59, 95% CI = 1.20–1.97, *p* < 0.001). Notably, even controlling for these structural characteristics of establishments, establishments with strong senior leadership support had a strong positive correlation with the number of wellness programs (β = 1.67, 95% CI = 1.25–2.10, *p* < 0.001).

When analyzing the adoption of specific workplace health programs, these findings became more nuanced. These results are found in Table 4, which presents logistic regression models to determine the likelihood of establishments having WHPs in nine different specific areas. As in the ordinary least squares analysis (Table 3), the smallest category of establishments (10–24 employees) and heavy industry were used in the base of the models.

Enterprises with more than 500 employees were more likely to have programs related to physical activity (OR = 2.41, 95% CI = 1.32–4.39, *p* < 0.01), nutrition (OR = 5.62, 95% CI = 2.97–10.64, *p* < 0.001), tobacco cessation (OR = 4.01, 95% CI = 2.25–7.16, *p* < 0.001), lactation support (OR = 4.94, 95% CI = 2.77–8.81, *p* < 0.001), and stress reduction (OR = 5.19, 95% CI = 2.76–9.76, *p* < 0.001). However, there was no statistically significant relationship between the largest category of enterprises and the likelihood of an establishment having WHPs to address obesity, alcohol abuse, MSD, and sleep. Establishments with 50–99 employees were less likely to have programs for obesity (OR = 0.56, 95% CI = 0.34–0.94, *p* < 0.05), alcohol abuse (OR = 0.53, 95% CI = 0.33–0.89, *p* < 0.05), and MSD (OR = 0.43, 95% CI = 0.25–0.75, *p* < 0.01). Establishments with 24–49 employees were less likely to have programs addressing MSD (OR = 0.61, 95% CI = 0.38–0.96, *p* < 0.05) and sleep (OR = 0.52, 95% CI = 0.31–0.87, *p* < 0.05).

When analyzing differences across industries, trade/retail was mostly indistinguishable from the agriculture/construction/mining/utilities category, which was the referent category in the base of the model. The one area of exception was that trade/retail establishments had a statistically significant higher likelihood of having a sleep program (OR = 1.97, 95% CI = 1.06–3.69, *p* < 0.05). Establishments in arts and foods services were more likely to have programs in obesity (OR = 2.06, 95% CI = 1.16–3.65, *p* < 0.05) and stress reduction (OR = 2.24, 95% CI = 1.28–3.29, *p* < 0.01) and less likely to have an MSD program (OR = 0.33, 95% CI: 0.17—0.64, *p* < 0.001). The info/tech/finance category was less likely to have programs to address tobacco (OR = 0.53, 95% CI = 0.31–0.91, *p* < 0.05), alcohol (OR = 0.43, 95% CI = 0.25–0.75, *p* < 0.01), and MSD (OR = 0.35, 95% CI = 0.19–0.63, *p* < 0.001). In education and healthcare category, there was a higher likelihood of establishments having a stress reduction program (OR = 1.72, 95% CI = 1.07–2.78, *p* < 0.05). This sector was less likely to have initiatives to address alcohol (OR = 0.44, 95% CI = 0.26–0.73, *p* < 0.001) and MSD risks (OR = 0.38, 95% CI = 0.22–0.64, *p* < 0.001). Establishments in public administration had a statistically significant higher likelihood of having programs to promote physical activities (OR = 2.22, 95% CI = 1.23–2.86, *p* < 0.01), nutrition (OR = 1.90, 95% CI = 1.08–3.33, *p* < 0.001, *p* < 0.001) and stress reduction (OR = 2.50, 95% CI = 1.42–4.40). Hospitals had strongest association with WHPs in five areas: physical activities (OR = 1.78, 95% CI = 1.01–3.16, *p* < 0.005), nutrition (OR = 1.92, 95% CI = 1.10–3.34, *p* < 0.005), obesity (OR = 2.12, 95% CI = 1.22–3.68, *p* < 0.01), lactation (OR = 3.14, 95% CI = 1.69–5.83, *p* < 0.001), and stress (OR = 2.33, 95% CI = 1.33–4.09, *p* < 0.01).

Controlling for the industrial sector and firm size, establishments in which at least 10 percent of the workforce were covered by a collective bargaining agreement had a higher likelihood of having all nine workplace health programs included in the study. These associations were at the *p <* 0.001 level of statistical significance, with the odds ratios ranging from OR = 1.66 for tobacco cessation to 4.05 for sleep programs. Full details of these statistical associations are provided in Table 4.

These strong effects were mirrored with the management Support variable, which had a consistently high odds ratio, with statistical significance noted at the *p* < 0.01 level for lactation programs and at the *p <* 0.001 level for the other eight programs. The odds ratios for this variable ranged from 2.00 for lactation programs to 4.62 for MSD programs.

Table 5 provides models of employee participation data in six areas for which we had sufficient data. In these models, the dependent variable equaled 1 if an establishment had more than 25 percent of its employees participating in the programs offered and zero for less than 25 percent As with the earlier tables, the smallest category of enterprises (10–24 employees) and heavy industry were considered in the base of the models. Although the largest category of enterprises (500+ employees) was associated with the adoption of WHP initiatives, these same enterprises were less likely to have more than 25 percent of employees participating in these programs when compared with the smallest enterprises for five of the six WHPs examined. For the sixth program (sleep), establishments with 25–49 employees had a higher likelihood of having these programs (OR = 5.48, 95% CI: 1.63–18.45, *p* < 0.01 when compared with the smallest establishments. With regard to the industry categories, establishments in public administration had a lower likelihood of having a 25% employee participation rate for all programs except for obesity programs. Unionization was associated with higher participation rates with exercise programs (OR = 1.53, 95% CI 1.00–2.32, *p* < 0.05) and lower participation in sleep programs (OR = 0.34, 95% CI 0.14–0.83, *p* < 0.05). Finally, senior leadership was associated with higher participation in nutrition (OR = 2.66, 95% CI 1.23–4.38, *p* < 0.05) and obesity programs (OR = 3.36, 95% CI 1.03–12.97, *p* < 0.05).

## 4. Discussion

This analysis indicates that firm size, industrial sector, unionization status, and management support have some level of explanatory power in predicting WHP adoption and employee participation in these programs. In these models, union status and management support had consistently stronger associations than structural features of the establishments.

The functional explanation for the relationship between establishment size and WHPs is premised on the assumption that the primary driver of WHPs is employer concerns about containing the costs of employee illness and absences associated with health behaviors. This functional approach to explaining the adoption of WHPs is also consistent with the focus on the industrial sector. In this analysis, we indeed observed some evidence of this. Hospitals were more likely to have a higher number of WHP initiatives and were more likely to adopt specific programs. These worksites would seem to have both the facilities and internal expertise that would lower the effective costs of program implementation. This was consistent with the strong association found in the adoption of lactation programs OR = 4.94, 95% CI = 2.77–8.81, *p* < 0.001). Similarly, the higher likelihood of establishments in public administration having programs in nutrition and physical activity may be understood in terms of the attributes of the industrial sector [24]. The long tenure of employees in public administration may support the economic case for WHPs since these programs may have long downstream savings in terms of preventing or mitigating chronic illness through prevention.

There are a few reasons for the limited explanatory power of this type of functional approach. The first is that many of the anticipated returns from investments may not be realized. This is related to the long-term nature of cost savings through behavioral health changes. The costs of WHPs occur at the time of implementation, and savings, if any, are typically only realized decades later. Most American workers have numerous employers over a lifetime. This, the return on investment in the health of employees is not meaningful to an enterprise when these individuals are subsequently employed elsewhere. Moreover, recent high-quality studies cast some doubt on the economic returns of wellness programs. Evidence from one large-scale study with randomized assignment suggests that those who chose to participate in establishment-based WHPs were already engaged in a range of healthy activities [14].

To appreciate the dynamics of WHPs, this analysis highlights the value of examining factors beyond size and industry. The presence of collective bargaining agreement was strongly associated with the number of programs in an establishment, the adoption of all nine WHP initiatives examined, and the higher employee participation in physical activities. Unionized establishments in the United States have better wages and benefits than their non-union counterparts [25]. Even when unions may only represent hourly employees in an enterprise, the bargained benefits for the unionized employees are matched for all employees in the establishment [26]. Additionally, in recent decades, unions have been limited in their ability to negotiate for higher wages, with employers sometimes offering benefits aimed at reducing health costs through improved access to wellness programs. Since unionized establishments also tend to have higher tenure employees [25], the economic case can be made for downstream savings from WHPs, as is the case with public administration establishments.

Controlling for firm size, industry, and unionization status, establishment-level leadership support for WHPs was the strongest predictor of the number of programs adopted, the likelihood of adoption of the nine WHP initiatives examined, and participation in nutrition and obesity programs. Whether motivated by costs or a desire to promote employee engagement, or normative values embedded in the organization, high-profile leadership support is an important factor in determining whether an establishment adopts a WHP initiative. Even in the absence of evidence of an economic return on wellness programs, organizations may believe that such a program can work well in their own organization and that the benefits of such programs extend beyond a promised financial return. Indeed, early adaptors of wellness programs have successfully leveraged these as part of their employment brand [27].

## 5. Conclusions

This analysis provides insights from both a functional and strategic choice perspective. When considering factors associated with establishments having WHPs, the structural attributes of establishments were important. The general relationship between firm size and the likelihood of WHPs was consistent with economies of scale enjoyed by large establishments. This was also observed in industry patterns of adoption. From this functional perspective, public policy efforts designed to decrease the costs of such programs and increase access to them would likely lead to more establishments adopting WHPs.

At the same time, this research highlights the importance of senior leadership support not just for the launch of WHPs but also for employee participation. Given that WHPs may not provide quick economic returns, it may be that only strong internal champions can ensure the resilience of these programs, particularly during periods when these programs do not offer a return on investment. Broadly, these findings suggested that functional attributes of establishments may change the cost calculus of WHPs and that senior organizational leadership’s commitment to the ideal of wellness may be a deciding factor in adoption. In this respect, the decision to adopt WHPs is not greatly different from other strategic choices management can make in the realm of human resources [28].

In all models, a relatively low amount of total variation was found, indicating a number of missing variables. These included workforce demographics, financial performance, workplace relations, and ownership structure, factors that may be included in future research. As with other WHA surveys, the 2017 WHA was cross-sectional, designed to provide a snapshot of WHPs in the US economy. Although the low response rate raises a concern about a systematic bias between non-respondent and respondents, the general findings of program frequency were consistent with other studies [19,20,21], and the main focus of the study was differences among respondents.

In addition to the inherent limitations of all cross-sectional research, this study occurred before the pandemic of 2020–2021. Even as the American economy returns to full employment, the return to work for many will be different. For some segments of the US workforce, there may not be a full return to work, as many establishments will see some advantage to a partially or fully remote or hybrid workforce. The implications of this for workplace wellness programs are unknown, but what is known is that strategic decisions of leaders can shape these programs.

## Figures and Tables

**Table 1 healthcare-10-01216-t001:** WHA Sample by industry and establishment size (percentage).

Industry Categories	Number of Employees						
	10–24	24–49	50–99	100–249	250–499	500+	Total
Agr, Mining, Util, Construction	221	155	75	44	18	12	525
	(7.8)	(5.5)	(2.6)	(1.5)	(0.6)	(0.4)	(18.5)
Trade/Retail	173	66	34	21	4	13	311
	(6.1)	(2.3)	(1.2)	(0.7)	(0.1)	(0.5)	(10.9)
Food Service/Art	219	125	50	21	6	12	433
	(7.7)	(4.4)	(1.8)	(0.7)	(0.2)	(0.4)	(15.2)
Info/Fin/Real	250	107	36	20	8	8	429
	(8.8)	(3.8)	(1.3)	(0.7)	(0.3)	(0.3)	(15.1)
Educ/Healthcare	206	121	109	62	13	40	551
	(7.2)	(4.3)	(3.8)	(2.2)	(0.5)	(1.4)	(19.4)
Public Admin	76	60	40	35	19	26	256
	(2.7)	(2.1)	(1.4)	(1.2)	(0.7)	(0.9)	(9.0)
Hospital	30	21	21	60	63	143	338
	(1.1)	(0.7)	(0.7)	(2.1)	(2.2)	(5.0)	(11.9)
Total	1175	655	365	263	131	254	2843
	(41.3)	(23.0)	(12.8)	(9.3)	(4.6)	(8.9)	(100.0)

*n* = 2843.

**Table 2 healthcare-10-01216-t002:** Firms with WHPs in last 12 months (percentage).

Industry Categories	Number of Employees						
	10–24	25–49	50–99	100–249	250–499	500+	Total
Agr, Mining, Util, Construction	62	53	39	35	14	12	215
	(30)	(35.3)	(54.9)	(83.3)	(77.8)	(100)	(43)
Trade/Retail	68	34	19	14	4	10	149
	(40.7)	(53.1)	(59.4)	(66.7)	(100)	(76.9)	(49.5)
Food Services/Art	72	39	25	12	4	10	162
	(34.3)	(34.5)	(51)	(60)	(66.7)	(83.3)	(39.5)
Info/Fin/Real	91	41	23	11	7	8	181
	(37.1)	(40.6)	(65.7)	(57.9)	(87.5)	(100)	(43.5)
Edu/Healthcare	94	57	69	42	8	37	307
	(47.2)	(49.6)	(64.5)	(73.7)	(66.7)	(92.5)	(57.9)
Public Admi	55	43	28	28	16	24	194
	(74.3)	(72.9)	(73.7)	(82.4)	(88.9)	(96)	(78.2)
Hospital	22	14	12	45	53	130	276
	(73.3)	(70)	(57.1)	(76.3)	(86.9)	(90.9)	(82.6)
Total	464	281	215	187	106	231	1484
	(41)	(45.2)	(60.9)	(74.2)	(83.5)	(91.3)	(54.2)

*n* = 2843.

**Table 3 healthcare-10-01216-t003:** Ordinary least square based on number of programs with unstandardized coefficients (95% confidence intervals).

	Model 1A	Model 1B	Model 1C	Model 1D
Constant	3.33 (3.07–3.59)	3.18 (2.74–3.61) ***	3.10 (2.63–3.57) ***	1.66 (1.08–2.25) ***
25–49 Employees	−0.47 (−0.90–−0.05) *	−0.52 (−0.94–−0.09) *	−0.50 (−0.96–−0.04) *	−0.45 (−0.90–0.00) *
50–99 Employees	−0.50 (−0.98–−0.03) *	−0.52 (−0.99–−0.04) *	−0.75 (−1.28–−0.23) **	−0.72 (−1.23–−0.21) **
100–249 Employees	0.44 (−0.04–0.92)	0.25 (−0.24–0.73)	0.22 (−0.32–0.77)	0.17 (−0.35–0.70)
250–499 Employees	0.62 (0.02–1.21) *	0.26 (−0.36–0.88)	0.13 (−0.59–0.84)	0.13 (−0.56–0.82)
500+ Employees	1.69 (1.25–2.14)	1.31 (0.81–1.81) ***	1.66 (1.08–2.25) ***	1.73 (1.17–2.30) ***
Trade/Retail		0.79 (0.21–1.37) **	0.60 (−0.01–1.22)	0.63 (0.04–1.23) *
Food Service/Arts		−0.22 (−0.81–0.36)	0.00 (−0.63–0.62)	0.09 (−0.52–0.70)
Info, Finance, Realty		−0.34 (−0.90–0.22)	−0.26 (−0.86–0.33)	−0.22 (−0.80–0.35)
Edu/Healthcare		−0.08 (−0.57–0.42)	−0.22 (−0.76–0.32)	−0.17 (−0.69–0.35)
Pub Admin		0.92 (0.38–1.47) ***	0.61 (−0.01–1.23)	0.71 (0.11–1.31) *
Hospitals		0.78 (0.24–1.33) **	0.78 (0.17–1.39) *	0.80 (0.21–1.39) **
Unionization			1.69 (1.29–2.09) ***	1.59 (1.20–1.97) ***
Sr Leadership Support				1.67 (1.24–2.10) ***
Adjusted R-Sq	0.07	0.09	0.18	0.23
F statistic	19.57	12.82	18.11	22.34
*n*	1290	1290	909	909

* *p* < 0.05, ** *p* < 0.01, *** *p* < 0.001.

**Table 4 healthcare-10-01216-t004:** Logistic regression with odds ratios (95% confidence intervals).

	Physical Activity	Nutrition	Obesity	Tobacco	Alcohol	Lactation	MSD	Stress	Sleep
25–49 Employees	0.82 (0.55–1.22)	0.91 (0.61–1.35)	0.81 (0.53–1.25)	0.68 (0.45–1.03)	0.89 (0.59–1.36)	0.93 (0.53–1.62)	0.61 (0.38–0.96) *	0.82 (0.55–1.22)	0.52 (0.31–0.87) *
50–99 Employees	0.65 (0.42–1.03)	0.74 (0.47–1.17)	0.56 (0.34–0.94) *	0.63 (0.40–1.00)	0.54 (0.33–0.89) *	0.87 (0.47–1.63)	0.43 (0.25–0.75) **	0.70 (0.44–1.11)	0.59 (0.33–1.05)
100–249 Employees	1.52 (0.91–2.53)	1.31 (0.82–2.10)	1.06 (0.64–1.74)	1.12 (0.69–1.80)	0.95 (0.58–1.55)	1.62 (0.92–2.85)	0.79 (0.47–1.32)	1.16 (0.72–1.85)	0.48 (0.26–0.89) *
250–499 Employees	1.44 (0.73–2.86)	1.84 (0.96–3.51)	1.16 (0.60–2.24)	1.13 (0.61–2.10)	0.58 (0.29–1.13)	1.61 (0.79–3.29)	0.73 (0.38–1.41)	1.15 (0.61–2.14)	0.25 (0.10–0.58) ***
500+ Employees	2.41 (1.32–4.39) **	5.62 (2.97–1.64) ***	1.40 (0.84–2.35)	4.01 (2.25–7.16) ***	1.46 (0.86–2.47)	4.94 (2.77–8.81) ***	1.48 (0.86–2.54)	5.19 (2.76–9.76) ***	1.13 (0.64–2.01)
Trade/Retail	1.52 (0.89–2.60)	1.59 (0.93–2.70)	1.74 (0.99–3.07)	1.52 (0.89–2.59)	1.19 (0.69–2.04)	0.69 (0.29–1.60)	1.08 (0.62–1.89)	1.52 (0.88–2.61)	1.97 (1.06–3.69) *
Food Service/Arts	1.54 (0.89–2.60)	1.70 (0.98–2.93)	2.06 (1.16–3.65) *	1.10 (0.64–1.90)	1.23 (0.71–2.14)	1.15 (0.51–2.55)	0.33 (0.17–0.64) ***	2.24 (1.28–3.92) **	0.61 (0.29–1.30)
Info/Fin/Real	1.29 (0.78–2.15)	1.13 (0.68–1.88)	0.91 (0.52–1.60)	0.53 (0.31–0.91) *	0.43 (0.25–0.75) **	1.65 (0.87–3.14)	0.35 (0.19–0.63) ***	1.18 (0.70–1.99)	1.00 (0.52–1.92)
Edu/Healthcare	1.40 (0.87–2.25)	1.37 (0.86–2.19)	0.83 (0.50–1.40)	0.67 (0.42–1.08)	0.44 (0.26–0.73) ***	1.58 (0.87–2.86)	0.38 (0.22–0.64) ***	1.72 (1.07–2.78) *	0.67 (0.36–1.23)
Public Admin	2.22 (1.23–4.01) **	1.90 (1.08–3.33) *	1.36 (0.77–2.41)	1.61 (0.93–2.79)	0.95 (0.55–1.66)	1.39 (0.70–2.77)	0.77 (0.44–1.36)	2.50 (1.42–4.40) **	1.24 (0.66–2.34)
Hospital	1.78 (1.01–3.16) *	1.92 (1.10–3.34) *	2.12 (1.22–3.68) **	1.55 (0.90–2.68)	0.70 (0.40–1.22)	3.14 (1.69–5.83) ***	0.82 (0.47–1.43)	2.33 (1.33–4.09) **	1.82 (0.95–3.47)
Union	1.93 (1.31–2.86) ***	2.78 (1.90–4.05) ***	2.81 (1.94–4.06) ***	1.66 (1.16–2.38) **	2.82 (1.96–4.06) ***	1.59 (1.04–2.44) *	2.93 (2.02–4.24) ***	2.52 (1.73–3.66) ***	4.05 (2.74–5.98) ***
Leadership Support	2.54 (1.74–3.71) ***	2.94 (1.96–4.39) ***	3.15 (2.01–4.94) ***	3.25 (2.13–4.98) ***	3.20 (2.02–5.06) ***	2.00 (1.20–3.35) **	4.62 (2.69–7.92) ***	3.14 (2.08–4.75) ***	3.73 (2.01–6.91) ***
Nagelkerke R-Sq *n*	0.125 884	0.205 892	0.165 807	0.199 864	0.171 834	0.217 742	0.220 831	0.213 868	0.219 828

* *p* < 0.05, ** *p* < 0.01, *** *p* > 0.001.

**Table 5 healthcare-10-01216-t005:** Logistic regression with odds ratios of 25% employee participation (95% confidence intervals).

	Exercise	Nutrition	Obesity	MSD	Stress	Sleep
25–49 Employees	0.48 (0.28–0.84) **	0.79 (0.42–1.51)	0.79 (0.30–20.4)	0.61 (0.26–1.40)	1.31 (0.68–2.52)	5.48 (1.63–18.45) **
50–99 Employees	1.10 (0.59–2.05)	1.38 (0.66–2.88)	1.69 (0.70–4.04)	1.03 (0.37–2.90)	1.00 (0.46–2.15)	1.33 (0.33–5.39)
100–249 Employees	0.57 (0.32–1.01)	0.46 (0.24–0.89) *	0.41 (0.15–1.12)	0.38 (0.15–0.95) *	0.88 (0.42–1.81)	3.15 (0.84–11.84)
250–499 Employees	0.62 (0.30–1.30)	0.38 (0.17–0.89) *	1.10 (0.39–3.07)	1.31 (0.36–4.82)	0.84 (0.31–2.25)	1.20 (0.18–8.19)
500+ Employees	0.32 (0.18–0.59) ***	0.31 (0.16–0.61) ***	0.66 (0.29–1.50)	0.19 (0.07–0.49) ***	0.34 (0.16–0.72) **	0.18 (0.03–1.03)
Trade/Retail	1.90 (0.94–3.86)	1.87 (0.84–4.19)	0.22 (0.06–0.77) *	1.74 (0.65–4.68)	2.05 (0.82–5.11)	0.49 (0.11–2.11)
Arts/Food Services	0.63 (0.31–1.29)	1.72 (0.67–4.38)	1.12 (0.33–3.76)	1.33 (0.40–4.41)	0.49 (0.20–1.20)	0.24 (0.03–1.89)
Info/Fin/Real	1.61 (0.81–3.20)	1.40 (0.62–3.17)	1.24 (0.44–3.47)	0.78 (0.26–2.30)	0.79 (0.31–2.01)	0.98 (0.20–4.79)
Edu/Healthcare	0.71 (0.38–1.32)	1.03 (0.52–2.07)	0.91 (0.37–2.24)	0.47 (0.19–1.20)	0.83 (0.38–1.81)	0.83 (0.19–3.64)
Publican Admin	0.50 (0.25–1.00) *	0.43 (0.19–0.94) *	0.43 (0.14–1.26)	0.22 (0.08–0.55) **	0.35 (0.14–0.85) *	0.15 (0.03–0.83) *
Hospitals	1.29 (0.67–2.49)	1.50 (0.71–3.16)	1.35 (0.56–3.29)	1.12 (0.43–2.92)	0.74 (0.32–1.71)	0.59 (0.14–2.55)
Unionization	1.52 (1.00–2.32) *	1.37 (0.88–2.14)	1.03 (0.58–1.84)	0.87 (0.47–1.58)	1.33 (0.80–2.20)	0.34 (0.14–0.83) *
Sr Leadership Support	2.41 (1.32–4.38)	2.66 (1.23–5.71) *	3.66 (1.03–12.97) *	1.98 (0.61–6.48)	2.22 (0.92–5.32)	1.15 (0.17–7.66)
Nagelkerke R-Sq	0.132	0.148	0.108	0.228	0.145	0.332
*n*	546	436	334	260	391	167

* *p* < 0.05, ** *p* < 0.01, *** *p* > 0.001.

## Data Availability

Data available from https://www.cdc.gov/workplacehealthpromotion/data-surveillance/index.html (accessed on 28 August 2021).

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
