# Peer review of "Strategic Choice and Implementation of Workplace Wellness Programs in the United States"

_healthcare, 2022, doi:10.3390/healthcare10071216_

Round 1
Reviewer 1 Report
Both linguistically and scholarly,the authors prepared an excellently designed analysis of an intervention to advance the health of aging workers.However,there are sentences that require copy editing throughout the earlier pages of the manuscript that require editing before the article is published in this or another journal.
Author Response
Thank you for your comments and time to review the manuscript. In our re-write we have been careful to copyedit the entire manuscript.
Reviewer 2 Report
This article starts from the shortage of health promotion programs in the US and tries to identify the determining factors. The results are interesting and could be useful for changing the trend.
1. In Table 3 the legend should be sufficiently informative, clearly explaining what it indicates, what is the meaning of the numbers displayed (probably correlation coefficients with confidence interval)
2. The regression models shown in Table 3 have a low coefficient of determination. Authors should discuss this point. Authors should accept that the proposed models explain a very modest portion of the total variance, at most 23%.
3. Probably other variables that did not fit into the model could have explained a larger portion of the variance. If the authors have any hypotheses about what these variables might be, this could be a starting point for further studies.
4. The same considerations on the clarity of the legend and on the low determination coefficient can be made for the data reported in Tables 4 and 5
Author Response
Thank you for your review of the manuscript. We have made a number of revisions, per your suggestion and the suggestion of other reviewers. Specifically, please note:
- New legends have been added to tables 3, 4, and 5
- Limitations related to the low coefficient of determination have been noted in the conclusion.
- We have noted possible missing variables
Once again, many thanks for your time and review.
Reviewer 3 Report
1. Summary
This paper deals with the workplace health programs in the US. The analysis provides a close look at the correlations that favor the adoption of health programs. By selecting a number of establishments and conducting interviews with some representatives, the author tries to explain the success factors for the implementation of such programs. The responses were then analyzed following two different methodologies: least square regression and logistic regression. The research shows that the most relevant factors in adopting WHPs are firm size, industrial sector, unionization status and management support. These findings appear in line with the current knowledge and extant literature, even though the paper did not provide references on this. One of the main strengths of the paper is the completeness of the analysis, which takes into consideration all enterprises included in the WHA survey. Despite the low response ratio, the large number of interviews is statistically significant.
2. Major issues
The literature review is totally focused on the need for establishing WHPs in the USA; there were no references to literature that could be compared with the present research, i.e. the factors that impact on the adoption of such provisions. Is there other work that try to link the diffusion of those programs to some structural factors? Some data and results could be better presented by using graphs and charts. This would make the outcomes easier to interpret.
3. Opinion
The work is well written, clear and concise.. The scope of the analysis is wide and data collected are statistically significant. The statistical approach is detailed, however the main methodological choices require more explanations. Literature review should be better focused on the aim of the research, i.e.: the factors that impact on WHP adoption. The results and conclusions are quite obvious.
Line 45 – Explain the acronym BMI
Line 64 – add references for the traditional organizational studies
Line 94 to 99: here is not clear how the sample is chosen
Line 100 – why did the authors choose the interview as methodology? A justification for that choice should be given
Line 108 – missing a which before ‘is low’
Line 110 – 116: this part deals with the uncertainty related to the sources chosen and should be placed in a proper section about uncertainty.
Author Response
Thank you for your time and thoughtful review. Per your recommendations, we have made the following revisions:
- Discussion of previous research exploring adoption of workplace health
- BMI is spelled out
- Revised wording on sample to explain that this is a sample of entire population
- Discussion of limitations included in conclusion
- Explanation of interview - interviews were used as screening to determine establishment eligibility
- Copy edited throughout
Once again, that you for your thought comments and your to review this manuscript.
This manuscript is a resubmission of an earlier submission. The following is a list of the peer review reports and author responses from that submission.
Round 1
Reviewer 1 Report
It is recommended to justify the study of a public survey of health conditions of individual employees - workers in various industrial sectors by, first analyzing the strengths and weaknesses of the Program. For instance, it (the Program) might appear to be a reporting mechanism instead of an obligatory device imposing duties or responsibilities on either employer or employee. Along the same reasoning, it has led to doubts when the manuscript provided no clues on the intentions and details of the Program by those health indicators listed in Table 4, which gave no ideas about who have been empowered to do what in the first place. On a contextual issue, it is unclear how an individual's health condition become the responsibility of either the employer or union. At it is now, it makes understanding and learning from the manuscript difficult since it offered no explanation of What, Why and How.
Author Response
Thank you for your insights. Additional comments attached.

Reviewer 2 Report
Dear Author
The article “Strategic Choice and Implementation of Workplace Wellness 2 Programs in the United States” is very important for the scientific community and more specifically for policies associated with the workplace.
The article is well organized with scientific and methodological rigor but suggested some changes/clarifications
1-line 92, (Materials and Methods) does not explain how the training of the investigator who collected the data. Must explain
2-line 152 and line 219 (results) where 25-49 refers does not match the content of tables 3 and 4- Must correct
3- The conclusions do not reflect the entirety of the study. The conclusions should work better and the limitations of the study should be identified.
Best Regards
Author Response

(The authors gave the same response as above.)

Reviewer 3 Report
This paper explores the importance of establishment characteristics, unionization, and strategic choice in the adoption of workplace health initiatives and employee participation in those initiatives. The research is based on data from the 2017 Workplace Health Administration Survey.
The paper is generally well-organized and presents a meaningful methodological tool along with very significant implications for the field of health policy. At the same time, this work does not provide a sufficient theoretical contribution. The background information provided in the literature review should be more extensive. Central concepts such as strategic choice are not sufficiently clear and are touched on only briefly. It would be good to expand this aspect of the paper and then to flesh out the conclusions and the researcher’s recommendations accordingly. Similarly, the discussion of the research literature is also insufficiently broad and some of the cited literature is out of date.
In conclusion, the paper presents a very important issue, but it would be appropriate for some corrections to be made (as noted above) before its publication in this journal.
Wishing you great success!

Author Response

(The authors gave the same response as above.)

Round 2
Reviewer 1 Report
Thanks for the revision unfortunately it has not addressed the comments made.
